# Late Myocardial Infarction and Repeat Revascularization after Coronary Artery Bypass Grafting in Patients with Prior Percutaneous Coronary Intervention [note 1]

**DOI:** 10.3390/jcm11195755

**Published:** 2022-09-28

**Authors:** Fausto Biancari, Antonio Salsano, Francesco Santini, Marisa De Feo, Magnus Dalén, Qiyao Zhang, Giuseppe Gatti, Enzo Mazzaro, Ilaria Franzese, Ciro Bancone, Marco Zanobini, Tuomas Tauriainen, Timo Mäkikallio, Matteo Saccocci, Alessandra Francica, Stefano Rosato, Zein El-Dean, Francesco Onorati, Giovanni Mariscalco

**Affiliations:** 1Heart and Lung Center, Helsinki University Hospital, University of Helsinki, 00029 Helsinki, Finland; 2Department of Medicine, South Karelia Central Hospital, University of Helsinki, 53130 Lappeenranta, Finland; 3Department of Surgery, Oulu University Hospital, University of Oulu, 90570 Oulu, Finland; 4Division of Cardiac Surgery, Ospedale Policlinico San Marino, University of Genoa, 16126 Genoa, Italy; 5Department of Cardiothoracic Sciences, Monaldi Hospital, University of Campania “Luigi Vanvitelli”, 81100 Naples, Italy; 6Department of Cardiac Surgery, Department of Molecular Medicine and Surgery, Karolinska Institutet, Karolinska University Hospital, 171 64 Solna, Stockholm, Sweden; 7Division of Cardiac Surgery, Cardio-Thoracic and Vascular Department, Azienda Sanitaria Universitaria Giuliano Isontina, 34148 Trieste, Italy; 8Cardiovascular Department, IRCCS Centro Cardiologico Monzino, 20138 Milan, Italy; 9Cardiac Surgery Unit, Poliambulanza Foundation, 25214 Brescia, Italy; 10Division of Cardiac Surgery, University of Verona Medical School, 37129 Verona, Italy; 11Istituto Superiore di Sanitá, 00161 Rome, Italy; 12Department of Cardiovascular Sciences, Clinical Sciences Wing, Glenfield Hospital, University of Leicester, Leicester LE3 9QP, UK

**Keywords:** coronary artery bypass grafting, percutaneous coronary intervention, myocardial infarction, repeat revascularization, prior PCI, previous PCI

## Abstract

Objectives: The aim of the present study was to evaluate the risk of late mortality and major adverse cardiovascular and cerebral events after coronary artery bypass grafting (CABG) in patients with prior percutaneous coronary intervention (PCI). Methods: A total of 2948 patients undergoing isolated CABGs were included in a prospective multicenter registry. Outcomes were adjusted for multiple covariates in logistic regression, Cox proportional hazards analysis and competing risk analysis. Results: In all, 2619 patients fulfilled the inclusion criteria of this analysis. Of them, 2199 (79.1%) had no history of PCI and 420 (20.9%) had a prior PCI. An adjusted analysis showed that a single prior PCI and multiple prior PCIs did not increase the risk of 30-day and 5-year mortality. Patients with multiple prior PCIs had a significantly higher risk of 5-year myocardial infarction (SHR 2.566, 95%CI 1.379–4.312) and repeat revascularization (SHR 1.774, 95%CI 1.140–2.763). Similarly, 30-day and 5-year mortality were not significantly increased in patients with prior PCI treatment of single or multiple vessels. Patients with multiple vessels treated with PCI had a significantly higher risk of 5-year myocardial infarction (SHR 2.640, 95%CI 1.497–4.658), repeat revascularization (SHR 1.648, 95%CI 1.029–2.638) and stroke (SHR 2.215, 95%CI 1.056–4.646) at 5-year. The risk for repeat revascularization was also increased with a prior single vessel PCI, but not for other outcomes. Conclusions: Among patients undergoing CABGs, multiple prior PCIs seem to increase the risk of late myocardial infarction and the need for repeat revascularization, but not the risk of mortality.

## 1. Introduction

Percutaneous coronary intervention (PCI) is the prevailing invasive treatment for coronary artery disease. In subsets of patients, PCI is associated with an increased risk of repeat revascularization compared to coronary artery bypass grafting (CABG) [1,2]. It is controversial whether a prior PCI may have a negative prognostic impact on the outcome of patients later requiring surgical myocardial revascularization. A recent pooled analysis demonstrated an increased risk of early mortality and major adverse cardiovascular events in patients with a prior PCI compared to those without a prior PCI [3]. However, this meta-analysis also included studies considering patients who underwent CABG shortly after PCI [4,5]. This might introduce bias because, in such cases, a CABG might have been performed for acute complications of a PCI or severe myocardial ischemia after an incomplete or unsuccessful PCI. In fact, large studies failed to detect an increased risk of early and late mortality after CABG when patients who underwent the operation after PCI during the index hospitalization or with recent acute coronary syndrome were excluded [6,7]. While studies focused mostly on the early and late mortality of these patients, data on major cardiovascular and cerebral events (MACCE) occurring after CABG for patients with prior PCI are scarce [3]. The aim of the present study is to evaluate the risk of late mortality and MACCE after CABG in patients with prior PCI from a multicenter registry. 

## 2. Material and Methods

### 2.1. Patient Population and Data Collection

The European Coronary Artery Bypass Grafting (E-CABG) registry was a prospective, multicenter study that included 7352 patients who underwent an isolated CABG at 16 European centers of cardiac surgery, which provided information for the evaluation of early outcomes after surgery from January 2015 to May 2017. The project is registered on ClinicalTrials.gov (Identifier: NCT02319083). Data on preoperative, operative and early postoperative variables and outcomes were prospectively collected. 

Eight centers (Genoa, Italy; Leicester, UK; Milan, Italy; Naples, Italy; Oulu, Finland; Stockholm, Sweden; Trieste, Italy; Verona, Italy) agreed to collect retrospective data on the late events of these patients. For the present study, only patients with complete data on the timing and number of PCIs and treated vessels, as well as data on the SYNTAX score, were included in this analysis. Patients who underwent a PCI ≤ 30 days from a CABG were excluded from this study because the indication for surgical revascularization might have been related to an acute complication of the PCI and/or severe myocardial ischemia not successfully treated by PCI. Data on the date of death, myocardial infarction, repeat coronary revascularization and stroke were collected retrospectively from electronic national registries as well as by contacting regional hospitals, patients and their relatives. The Ethical Committee of the sponsor institution granted the permission to perform this study (Ethic Committee Name: Pohjois-Pohjanmaan Sairaanhoitopiiri, Finland, approval Code: 195/2014; approval date: 20 October 2014).

The Institutional Review Board or Ethical Committee of each participating center has further approved this study.

### 2.2. Study Cohorts

A previous study [8] suggested that the impact of a prior PCI on late outcomes after CABG may be related to the number of prior PCI procedures. Therefore, in this study, we evaluated the outcomes of these patients based on the number of prior PCIs as well as the number of vessels treated with each PCI. The study cohorts of the first analysis were categorized as follows: (1) no prior PCI; (2) 1 prior PCI, i.e., single prior PCI; (3) ≥2 prior PCIs, i.e., multiple prior PCIs. The study cohorts of the second analysis were categorized as follows: (1) no prior PCI; (2) 1 vessel treated with prior PCI, i.e., single vessel PCI; (3) ≥2 vessels treated with prior PCIs, i.e., multiple vessel PCIs.

### 2.3. Outcomes

The primary outcomes were 5-year all-cause mortality, myocardial infarction, stroke and repeat coronary revascularization. Secondary outcomes were 30-day mortality as well as a composite of major cardiac and cerebrovascular events (MACCE), i.e., all-cause mortality, myocardial infarction, stroke and repeat coronary revascularization. For the present study, myocardial infarction was defined as any myocardial infarction occurring 30 days after the index CABG procedure because the diagnosis of perioperative myocardial infarction might have differed between centers. Repeat revascularization and stroke were considered when occurring at any time after surgery.

### 2.4. Statistical Analysis

The Kruskal–Wallis test and the linear-by-linear association test were used for univariate analysis of the baseline and operative data of the study cohorts. The Kaplan–Meier method and the Cox proportional hazard method were employed to evaluate the impact of the number of prior PCIs and the number of the main coronary vessels treated by a PCI on mortality and any MACCE. Since myocardial infarction, stroke and repeat coronary revascularization may be hindered by a patient’s death, a competing risk analysis using the Fine–Gray regression model with all-cause death as a competing event was performed to estimate the unadjusted and adjusted subdistribution hazard ratios (SHRs) for the incidence of these events. Multivariable models assessing the outcomes in patients with an increasing number of PCIs and an increasing number of treated main coronary arteries by PCIs were adjusted risk factors with a *p*-value < 0.20 for between-groups differences. Analyses on the impact of the number of prior PCIs as well as the impact of the number of vessels treated by prior PCIs were adjusted for the following covariates: age, dialysis, diabetes, atrial fibrillation, prior cardiac surgery, critical preoperative state, off-pump surgery, bilateral internal mammary artery grafting, radial artery grafting, number of diseased vessels, Synergy between Percutaneous Coronary Intervention with Taxus and Cardiac Sur-gery (SYNTAX) score and EuroSCORE II. The number of distal anastomoses was not included in the regression models because this might have reflected the number of vessels previously treated with PCIs. Indeed, revascularization can be considered complete in cases when previously stented vessels were free of signs of restenosis and therefore were not surgically revascularized. Cardiopulmonary bypass time and aortic cross-clamping time were not included in the regression models because a significant proportion of patients underwent revascularization with the off-pump technique. All statistical tests were two-sided, and *p* < 0.05 was set for the statistical significance. Statistical analyses were performed using SPSS v. 25 (IBM Corporation, New York, Armonk, NY, USA) and Stata v. 15.1 (StataCorp LLC, College Station, TX, USA) statistical softwares.

## 3. Results

Patient selection for this study is shown in Figure 1.

Out of 2948 patients who were operated on at the participating institutions from January 2015 to May 2017 and who were included in the E-CABG registry, 2619 patients fulfilled the inclusion criteria of this analysis. Of them, 2199 (79.1%) had no history of PCI and 420 (20.9%) had a prior PCI. A single PCI procedure was previously performed in 272 patients and ≥2 prior PCIs in 148 patients. A PCI procedure was previously performed on a single vessel in 271 patients and on ≥2 vessels in 149 patients. The mean follow-up time of this series was 4.6 ± 1.3 years. The mean time for myocardial infarction was 2.6 ± 1.5 years, for repeat revascularization 1.7 ± 1.7 years and for stroke 1.8 ± 1.8 years.

### 3.1. Outcome According to the Number of Prior PCIs

Patients’ characteristics in these study groups are summarized in Table 1. Patients with multiple PCIs had an increased proportion of in-stent restenosis (53.1% vs. 22.1%) and stent thrombosis (5.4% vs. 2.2%) compared to patients with a single prior PCI. Besides a higher operative risk, patients with multiple prior PCIs had a lower SYNTAX score compared with the other study groups. Patients with prior PCIs had significantly less distal anastomoses, particularly among those with multiple prior PCIs.

Patients with a single prior PCI or multiple prior PCIs did not have an increased risk of 30-day mortality when adjusted for multiple covariates (Table 2). Similarly, 5-year mortality and stroke were not significantly increased in patients with a prior PCI compared to those without a prior PCI. However, patients with multiple prior PCIs had a significantly higher risk of late myocardial infarction (SHR 2.566, 95%CI 1.379−4.312) and repeat revascularization (SHR 1.774, 95%CI 1.140−2.763) at 5 years (Figure 2). An increased risk for repeat revascularization was also observed among those patients with a single prior PCI (SHR 1.550, 95%CI 1.049−2.291). 

### 3.2. Outcome According to the Number of Treated Vessels with PCIs

Patients’ characteristics are summarized in Table 3. Patients with multiple vessel PCIs had a markedly increased proportion of in-stent restenosis (50.0.1% vs. 23.6%) compared to patients with one prior PCI. Patients with multiple vessel PCIs had a significantly increased operative risk as estimated by the EuroSCORE II. However, patients with multiple vessel PCIs had lower (SYNTAX) scores compared with the other study groups, as well as a lower mean number of diseased vessels. Patients with a prior PCI had significantly less distal anastomoses, particularly among those with multiple vessel PCIs.

Patients with a single vessel treated with a PCI and those with multiple vessels treated with PCIs did not have a significantly increased risk of 30-day mortality when adjusted for multiple covariates (Table 4). Similarly, 5-year mortality was not significantly increased in patients with a prior PCI compared to those without a prior PCI. However, patients with multiple vessels treated with PCIs had a significantly higher risk of late myocardial infarction (SHR 2.640, 95%CI 1.497−4.658), repeat revascularization (SHR 1.648, 95%CI 1.029−2.638), stroke (SHR 2.215, 95%CI 1.056−4.646) and MACCE (HR 1.655, 95%CI 1.211−2.260) at 5 years. An increased risk for repeat revascularization was also observed among patients with a single vessel treated by a prior PCI (SHR 1.627, 95%CI 1.115−2.373). 

## 4. Discussion

The present multicenter study demonstrated that having multiple prior PCIs may increase the risk of myocardial infarction and repeat revascularization after CABG at a 5-year follow-up. The novelty of this analysis resides in the information provided on major adverse cardiovascular events at 5-year follow-up in a rather large series of all-comers populations. In fact, there is some evidence from several studies that a prior PCI might not affect late survival after a CABG [3], but, to the best of our knowledge, only a few studies reported on the adjusted estimates of late non-fatal events after CABGs in patients with a prior PCI [8,9,10,11]. These studies did not employ competing risk methods, and non-fatal ischemic events were poorly defined. Similarly, only a very few studies performed multivariable analyses on the impact of a prior PCI on late mortality [6,12,13]. Hakamada et al. [8] were the only investigators to perform an analysis on multiple late adverse events, i.e., all-cause mortality, cardiac-related mortality, myocardial infarction, and repeated revascularization, considering prior single and multiple PCIs. Although these authors did not consider a patient’s death as a competing risk, they demonstrated that only patients who had multiple PCIs had an increased risk of 10-year all-cause mortality, cardiac-related mortality, and myocardial infarction, with a trend toward an increased risk of repeat revascularization as well [8]. In Hakamada’s study [8], patients with a single prior PCI did not have any increased risk of late adverse events. To some extent, our results confirmed that multiple prior PCIs have a negative prognostic impact in terms of increased risk of repeat revascularization and myocardial infarction. Patients with a single prior PCI did not have an increased risk of adverse events, with the exception of an increased risk of repeat revascularization. It is worth noting that our analysis showed that the negative prognostic impact of prior PCIs seemed more evident in patients who underwent PCIs on multiple vessels.

Taken together, the present findings, along with the results by Hakamada et al. [8], suggest that a single prior PCI procedure is unlikely to jeopardize the late outcome of a CABG. By contrast, multiple PCI procedures, particularly if performed on multiple vessels, may reduce the benefits of CABGs in the long run by increasing the risk of myocardial infarction and the need for further revascularization procedures. The rather long follow-up in the study by Hakamada et al. [8] demonstrated that patients with multiple prior PCIs might also have a decreased expectancy of life. This significant effect of prior PCIs on late survival should be confirmed in studies with a long follow-up considering the impact of multiple PCI procedures and the number of treated vessels.

This study was not planned to address the causes of the increased risk of MACCE associated with a prior PCI. However, we speculate that multiple PCIs, as well as a PCI on multiple vessels, may prevent surgical revascularization on optimal sites of diseased coronary arteries, and coronary stents may not allow the development of collateral circulation. Furthermore, a failing stent may contribute to myocardial infarction and the need for repeat revascularization. Indeed, the present study showed that patients with prior PCIs, particularly those with multiple prior PCIs, had lower SYNTAX scores. This finding suggests that the extent of coronary artery disease per se should not be the main determinant of poor outcomes in these patients. By contrast, restenosis or even thrombosis of a previously implanted stent may contribute to some of such late coronary events. Cardiac surgeons planning CABGs often face questions about the durability of coronary stent, particularly in case of non-hemodynamically significant stent restenosis. In such cases, surgical revascularization might be considered incomplete. In fact, the prothrombotic state during the perioperative period, as well as the progression of coronary artery disease later, may further contribute to coronary stent failure and jeopardize the outcome of a CABG.

This study demonstrated a higher risk of early and late stroke in patients with ≥2 vessels treated with a prior PCI, and the same trend was also observed in patients with ≥2 prior PCIs. We speculate that the non-significantly increased baseline prevalence of stroke or transient ischemic attack, atrial fibrillation and left ventricular ejection fraction might have contributed to an overall increased risk of stroke of peripheral vascular and/or cardiogenic origin. When these three risk factors were included in the regression models, patients with ≥2 vessels treated with a prior PCI still had a significantly higher risk of developing stroke after a CABG (SHR 2.199, 95%CI 1.068–4.530). Still, a history of stroke or transient ischemic attack (SHR 2.226, 9%CI 1.191–4.159) and atrial fibrillation (SHR 1.941, 95%CI 1.085-3.461), but not a depressed left ventricular ejection fraction, were independent predictors of stroke. However, we do not have data on the nature of stroke and its underlying causes.

A few limitations might have affected the validity of the present findings. First, despite the prospective nature of the E-CABG registry, information on late events was retrospectively collected using different methods to gather this data. Second, we do not have information regarding the adverse events related to coronary stent or graft failure. Third, the fate of the previously implanted stents and atherosclerosis of native arteries might have been affected by the preoperative withdrawal of antiplatelet drugs and anticoagulants. Similarly, antithrombotic therapy might have had an impact on thrombotic events, early and late, after surgery [14]. However, collinearity between the use of antithrombotic drugs and prior PCIs prevents a reliable analysis of the impact of using these drugs on the outcomes. Finally, retrieval of information on mortality secondary to cardiac causes was not feasible in this study.

The characteristics of the participating centers were somewhat different in terms of patient volume, referral pathway, and revascularization strategy. Such inter-institutional differences make this study based on an all-comers population and strengthen the generalizability of these findings.

In conclusion, the results of the present study suggest that for patients undergoing a CABG, prior multiple PCIs may significantly increase the risk of late myocardial infarction and the need for repeat revascularization. Prior PCIs did not affect either early or late mortality after a CABG. These findings should be considered in the decision-making process, particularly when planning PCIs for complex coronary artery disease or when repeat revascularization after a prior PCI is indicated.

## Figures and Tables

**Figure 1 jcm-11-05755-f001:**
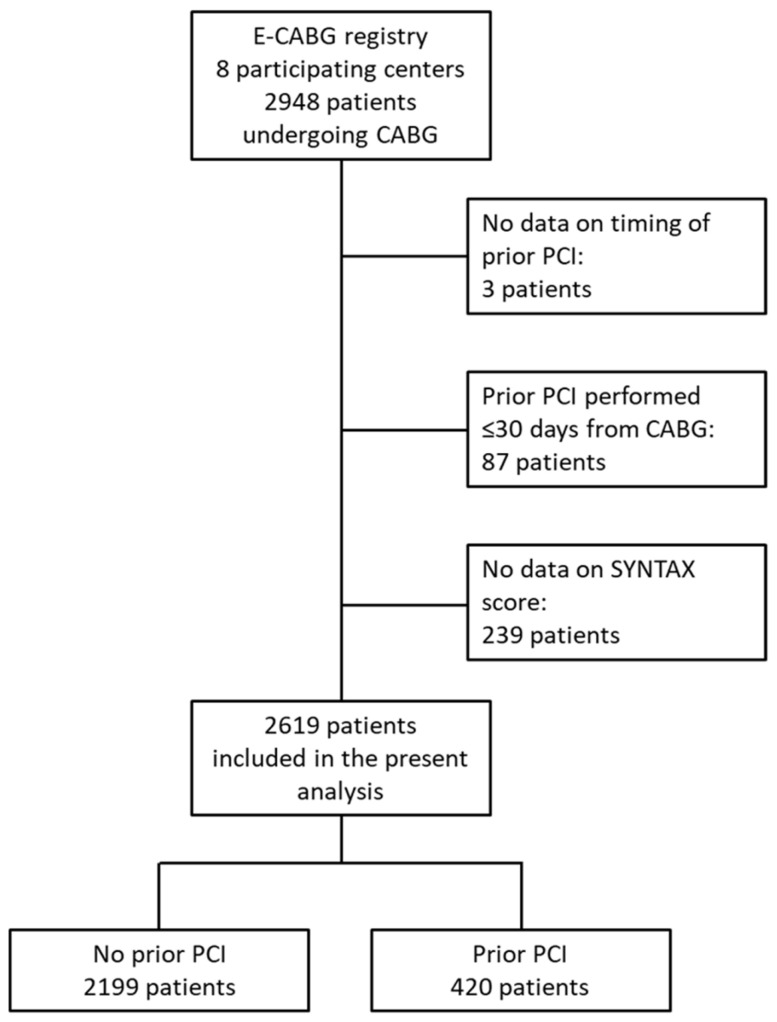
Flow chart of patient selection for the present analysis. CABG: coronary artery bypass grafting; PCI: percutaneous coronary intervention; SYNTAX: synergy between PCI with taxus and cardiac surgery.

**Figure 2 jcm-11-05755-f002:**
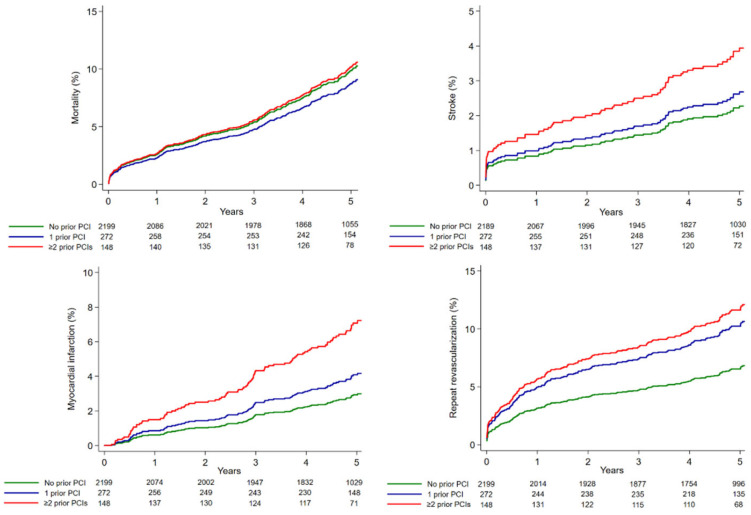
Adjusted estimates of outcomes in patients without prior PCI, with 1 prior PCI and with ≥2 prior PCIs.

**Table 1 jcm-11-05755-t001:** Baseline characteristics and operative data in the unmatched and matched cohorts.

Covariates	No Prior PCI 2199 pts	1 Prior PCI 272 pts	≥2 Prior PCIs 148 pts	*p*-Value
*Baseline risk factors*				
Age (years)	67.6 ± 9.2	65.6 ± 10.0	66.9 ± 8.8	0.006
Female	362 (16.5)	43 (15.8)	28 (18.9)	0.615
eGFR (mL/min/1.73 m^2^)	76 ± 20	76 ± 21	75 ± 23	0.807
Dialysis	19 (0.9)	3 (1.1)	5 (3.4)	0.009
Functioning kidney transplant	6 (0.3)	1 (0.4)	0 (0)	0.712
Anemia	607 (27.6)	67 (24.6)	49 (33.1)	0.507
Diabetes	632 (28.7)	87 (32.0)	54 (36.5)	0.028
Recent STEMI	140 (6.4)	14 (5.1)	10 (6.8)	0.812
Prior stroke/TIA	145 (6.6)	22 (8.1)	11 (7.4)	0.431
Atrial fibrillation	183 (8.3)	15 (5.5)	22 (14.9)	0.136
Pulmonary disease	241 (11.0)	31 (11.4)	17 (11.5)	0.785
Extracardiac arteriopathy	460 (20.9)	57 (21.0)	38 (25.7)	0.256
Prior cardiac surgery	7 (0.3)	4 (1.5)	5 (3.4)	<0.0001
LVEF ≤ 50%	666 (30.3)	81 (29.8)	47 (32)	0.793
Critical preoperative state	225 (10.2)	22 (8.1)	11 (7.4)	0.141
Emergency procedure	84 (3.8)	5 (1.8)	5 (3.4)	0.292
Indication for surgery				
In-stent restenosis	-	60 (22.1)	78 (53.1)	-
Stent thrombosis	-	6 (2.2)	8 (5.4)	-
Progression of CAD	-	211 (77.6)	118 (79.7)	-
Left main coronary a. PCI	-	8 (2.9)	15 (10.1)	-
Any drug-eluting stent	-	126 (46.3)	106 (71.6)	-
Delay from last PCI (years)	-	6.6 ± 6.4	4.5 ± 4.6	-
No. of diseased vessels	2.8 ± 0.5	2.6 ± 0.6	2.6 ± 0.6	<0.0001
Left main stenosis	213 (9.7)	25 (9.2)	11 (7.4)	0.381
SYNTAX score	29 ± 10	27 ± 10	26 ± 12	<0.0001
EuroSCORE II (%)	3.2 ± 4.5	3.0 ± 5.0	3.6 ± 4.6	0.065
*Operative data*				
No. of distal anastomoses	3.0 ± 1.0	2.7 ± 0.9	2.5 ± 0.9	<0.0001
CPB time (min)	89 ± 33	84 ± 32	88 ± 36	0.064
Aortic clamping time (min)	61 ± 27	58 ± 25	59 ± 29	0.041
Off-pump surgery	265 (12.1)	40 (14.7)	34 (23.0)	<0.0001
BIMA grafting	518 (23.6)	75 (27.6)	16 (10.8)	0.029
Radial artery graft	70 (3.2)	14 (5.1)	7 (4.7)	0.097

Continuous variables are reported as the mean ± standard deviation. Categorical variables are reported as counts and percentages. Anemia is defined as baseline hemoglobin concentration <12.0 g/L in women and <13.0 g/L in men. BIMA = bilateral internal mammary artery; CAD = coronary artery disease; CPB = cardiopulmonary bypass; eGFR = estimated glomerular filtration rate according to the CKD-EPI equation; EuroSCORE = European System for Cardiac Operative Risk Evaluation; LVEF = left ventricular ejection fraction; PCI = percutaneous coronary intervention; STEMI = ST-elevation myocardial infarction; TIA = transient ischemic attack. Clinical variables are according to the EuroSCORE II definition criteria.

**Table 2 jcm-11-05755-t002:** Crude rates and adjusted risk estimates of outcomes in the study cohorts.

	No Prior PCI 2199 pts	1 Prior PCI 272 pts	≥2 Prior PCIs 148 pts
30-day mortality	1.4%	1.8%	2.0%
Adjusted OR	-	1.504 (0.547−4.130)	1.325 (0.342−5.138)
5-year mortality	12.5%	9.3%	14.6%
Adjusted HR	-	0.867 (0.580−1.298)	0.993 (0.628−1.571)
5-year myocardial infarction	3.2%	5.3%	10.9%
Adjusted SHR	-	1.389 (0.752−2.680)	**2.566 (1.379−4.312)**
5-year repeat revascularization	7.2%	11.8%	15.3%
Adjusted SHR	-	**1.550 (1.049−2.291)**	**1.774 (1.140−2.763)**
5-year stroke	3.0%	3.0%	4.8%
Adjusted SHR	-	1.251 (0.596−2.727)	1.872 (0.874−4.001)
5-year MACCE	18.8%	20.5%	30.2%
Adjusted HR	-	1.188 (0.895−1.578)	1.383 (0.996−1.920)

PCI = percutaneous coronary intervention; MACCE = major adverse cardiac and cardiovascular events. OR = odds ratio; HR = hazard ratio; SHR = subdistributional hazard ratio.

**Table 3 jcm-11-05755-t003:** Baseline characteristics and operative data in the unmatched and matched cohorts.

Covariates	No Prior PCI 2199 pts	1 Vessel Treated with PCI 271 pts	≥2 Vessels Treated with PCI 149 pts	*p*-Value
*Baseline risk factors*				
Age (years)	67.6 ± 9.2	65.9 ± 9.9	66.4 ± 9.0	0.011
Female	362 (16.5)	44 (16.2)	27 (18.1)	0.698
eGFR (mL/min/1.73 m^2^)	76 ± 20	77 ± 21	74 ± 23	0.658
Dialysis	19 (0.9)	4 (1.5)	4 (2.7)	0.026
Functioning kidney transplant	6 (0.3)	1 (0.4)	0 (0)	0.711
Anemia	607 (27.6)	69 (25.5)	47 (31.5)	0.634
Diabetes	632 (28.7)	84 (31.0)	57 (38.3)	0.016
Recent STEMI	140 (6.4)	12 (4.4)	12 (8.1)	0.955
Prior stroke/TIA	145 (6.6)	19 (7.0)	14 (9.4)	0.225
Atrial fibrillation	183 (8.3)	18 (6.6)	19 (12.8)	0.279
Pulmonary disease	241 (11.0)	32 (11.8)	16 (10.7)	0.887
Extracardiac arteriopathy	460 (20.9)	53 (19.6)	42 (28.2)	0.141
Prior cardiac surgery	7 (0.3)	5 (1.8)	4 (2.7)	< 0.0001
LVEF ≤50%	666 (30.3)	79 (29.2)	49 (33.1)	0.691
Critical preoperative state	225 (10.2)	21 (7.7)	12 (8.1)	0.174
Emergency procedure	84 (3.8)	4 (1.5)	6 (4.1)	0.389
Indication for surgery				
In-stent restenosis	-	8 (3.0)	6 (4.0)	-
Stent thrombosis	-	64 (23.6)	74 (50.0)	-
Progression of CAD	-	205 (75.6)	124 (83.2)	-
Left main coronary a. PCI	-	0 (0)	23 (15.4)	-
Any drug-eluting stent	-	124 (45.8)	108 (72.5)	-
Delay from last PCI (years)	-	6.7 ± 6.4	4.4 ± 4.4	-
No. of diseased vessels	2.8 ± 0.5	2.6 ± 0.6	2.6 ± 0.6	<0.0001
Left main stenosis	213 (9.7)	19 (7.0)	17 (11.4)	0.891
SYNTAX score	29 ± 10	27 ± 10	26 ± 11	<0.0001
EuroSCORE II (%)	3.2 ± 4.5	3.0 ± 5.0	3.6 ± 4.6	0.035
*Operative data*				
No. of distal anastomoses	3.0 ± 1.0	2.7 ± 0.9	2.5 ± 0.9	<0.0001
CPB time (min)	89 ± 33	85 ± 32	86 ± 35	0.054
Aortic clamping time (min)	61 ± 27	59 ± 26	57 ± 27	0.053
Off-pump surgery	265 (12.1)	44 (16.2)	30 (20.1)	0.001
BIMA grafting	518 (23.6)	74 (27.3)	17 (11.4)	0.035
Radial artery graft	70 (3.2)	12 (4.4)	9 (6.0)	0.041

Continuous variables are reported as the mean ± standard deviation. Categorical variables are reported as counts and percentages. Anemia is defined as baseline hemoglobin concentration <12.0 g/L in women and <13.0 g/L in men. BIMA = bilateral internal mammary artery; CAD = coronary artery disease; CPB = cardiopulmonary bypass; eGFR = estimated glomerular filtration rate according to the CKD-EPI equation; EuroSCORE = European System for Cardiac Operative Risk Evaluation; LVEF = left ventricular ejection fraction; PCI = percutaneous coronary intervention; STEMI = ST-elevation myocardial infarction; TIA = transient ischemic attack. Clinical variables are according to the EuroSCORE II definition criteria.

**Table 4 jcm-11-05755-t004:** Crude rates and adjusted risk estimates of outcomes in the study cohorts.

	No prior PCI 2199 pts	1 Vessel Treated with PCI 271 pts	≥2 Vessels Treated with PCI 149 pts
30-day mortality	1.4%	1.5%	2.7%
Adjusted OR	-	1.165 (0.384−3.532)	1.937 (0.597−6.290)
5-year mortality	12.5%	6.9%	18.8%
Adjusted HR	-	0.641 (0.408−1.009)	1.383 (0.918−2.083)
5-year myocardial infarction	3.2%	5.3%	10.8%
Adjusted SHR	-	1.500 (0.831−2.705)	**2.640 (1.497−4.658)**
5-year repeat revascularization	7.0%	12.4%	14.1%
Adjusted SHR	-	**1.627 (1.115−2.373)**	**1.648 (1.029−2.638)**
5-year stroke	3.0%	2.6%	5.4%
Adjusted SHR	-	1.063 (0.489−2.310)	**2.215 (1.056−4.646)**
5-year MACCE	18.8%	18.5%	33.5%
Adjusted HR	-	1.041 (0.774−1.400)	**1.655 (1.211−2.260)**

PCI = percutaneous coronary intervention; MACCE = major adverse cardiac and cardiovascular events; OR = odds ratio; HR = hazard ratio; SHR = subdistributional hazard ratio.

## Data Availability

Not available because of privacy issues.

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
