# Peer review of "Late Myocardial Infarction and Repeat Revascularization after Coronary Artery Bypass Grafting in Patients with Prior Percutaneous Coronary Interventionâ€"

_jcm, 2022, doi:10.3390/jcm11195755_

Round 1
Reviewer 1 Report
In this study the authors retrospectively evaluated the long-term outcomes including MACCE after CABG in patients with prior PCI in comparison to those in patients without a PCI history.
Among the 2619 patients who fulfilled the inclusion criteria, 2199 (79.1%) had no history of PCI and 420 (20.9%) had a prior PCI. Multivariable analyses showed that single prior and multiple prior PCIs did not increase the risk of 30-day and 5-year mortality. Patients with multiple prior PCIs had a significantly higher risk of 5-year myocardial infarction and repeat revascularization. Similarly, 30-day and 5-year mortality were not significantly increased in patients with prior PCI treating single or multiple vessels. Patients with multiple vessels treated with PCI had a significantly higher risk of 5-year myocardial infarction, repeat revascularization, and stroke at 5-year. The risk for repeat revascularization was increased also with single vessel prior PCI, but not for other outcomes.
The authors concluded that multiple prior PCIs seem to increase the risk of late myocardial infarction and need of repeat revascularization, but not the risk of mortality.
This study sought to elucidate the impact of previous PCIs on early and late outcomes, including MACCE, after isolated CABG, excluding those performed for complications or failure of PCI. Compared to the 2 previous studies (Ref 6 & 7), this study has the strength that it evaluated long-term MACCE, and analyzed the outcomes according to the number of PCIs and the number of vessels involved. On the other hand, it has the weakness that it was a multi-center study (previous 2 studies were based on the national database), which may not reflect the real-world data because the participating centers maybe highly selected specialized centers. Compared to the Hakamata’s study, the present study has the merit of multicenter study, although the number of patients was comparable, no details of PCI were available, and the follow-up duration was shorter. The results were in line with those of previous studies, and add to the current knowledges of physicians.
Specific comments:
(Line 122-125) Why the number of distal anastomoses, CPB time, and aortic crossclamping time were not included in the 12 covariates used for multivariable models regarding the number of PCIs, although the p-values were less than 0.2?
(Statistical analysis) Covariates used for the models on the number of vessels involved were not specified.
(Table 1) What were the reasons for emergency surgery? These patients may have better been excluded.
What do the authors think was the reason for higher stroke rate in patients with prior PCIs for multi-vessel disease? Were they related to atrial fibrillation? Since this paper dealt with MACCE, not MACE, it may be better to be addressed briefly.
Author Response
1. This study sought to elucidate the impact of previous PCIs on early and late outcomes, including MACCE, after isolated CABG, excluding those performed for complications or failure of PCI. Compared to the 2 previous studies (Ref 6 & 7), this study has the strength that it evaluated long-term MACCE, and analyzed the outcomes according to the number of PCIs and the number of vessels involved. On the other hand, it has the weakness that it was a multi-center study (previous 2 studies were based on the national database), which may not reflect the real-world data because the participating centers maybe highly selected specialized centers. Compared to the Hakamata’s study, the present study has the merit of multicenter study, although the number of patients was comparable, no details of PCI were available, and the follow-up duration was shorter. The results were in line with those of previous studies, and add to the current knowledges of physicians.
Response: We are grateful to the Reviewer for her/his relevant comments on the nature of the study. We do believe that the characteristics of the participating centers were somewhat different in terms of patient’s volume, referral pathway and in the revascularization strategy. Such differences make this study based on the real-world data
Changes: We added the following comment to the Limitations section: “The characteristics of the participating centers were somewhat different in terms of patient’s volume, referral pathway and in the revascularization strategy. Such inter-institutional differences make this study based on the real-world data, which strengthens the value of this study”.
2. (Line 122-125) Why the number of distal anastomoses, CPB time, and aortic crossclamping time were not included in the 12 covariates used for multivariable models regarding the number of PCIs, although the p-values were less than 0.2?
Response: We thank this Reviewer for her/his detailed and critical review of our analyses. We did not include the number of distal anastomoses because this might have reflected the number of vessels previously treated with PCI. Indeed, revascularization can be considered complete in cases when previously stented vessels were free of signs of restenosis and therefore were not surgically revascularized. Still, we tested the impact of number of distal anastomoses as suggested by the Reviewer and this did not change the results (for ≥2 prior PCIs: late myocardial infarction, SHR 2.634, 95%CI 1.477-4.699; repeat revascularization, SHR 1.801, 95%CI 1.080-2.676; for ). Cardiopulmonary bypass time and aortic cross-clamping time were not included in the regression model because a significant proportion of patients underwent revascularization with the off-pump technique.
Changes: We added the following sentences to the Methods section: “The number of distal anastomoses was not included into regression models because this might have reflected the number of vessels previously treated with PCI. Indeed, revascularization can be considered complete in cases when previously stented vessels were free of signs of restenosis and therefore were not surgically revascularized. Cardiopulmonary bypass time and aortic cross-clamping time were not included in the regression model because a significant proportion of patients underwent revascularization with the off-pump technique.”.
3. (Statistical analysis) Covariates used for the models on the number of vessels involved were not specified.
Response: We do agree with the Reviewer and we added this information to the Methods section.
Changes: We added the information that the same covariates included into regression model for the number of prior PCIs as for the number of vessels treated by PCI.
4. What were the reasons for emergency surgery? These patients may have better been excluded.
Response: We partially agree with the Reviewer on this issue, because excluding patients undergoing surgery for acute coronary syndrome (some requiring emergency procedure) might have prevented an overall analysis of the impact of prior PCIs in a “real-world” setting.
Changes: None.
5. What do the authors think was the reason for higher stroke rate in patients with prior PCIs for multi-vessel disease? Were they related to atrial fibrillation? Since this paper dealt with MACCE, not MACE, it may be better to be addressed briefly.
Response: We fully agree on this issue, and we performed additional analysis and added comments on this issue to the manuscript.
Changes: We added this comment on this issue to the Discussion: “This study demonstrated a higher risk of early and late stroke in patients with ≥2 vessels treated with prior PCI and the same trend was observed also in patients with ≥2 prior PCIs. We speculate that the non-significantly increased baseline prevalence of stroke or transient ischemic attack, atrial fibrillation and left ventricular ejection fraction might have contributed to an overall increased risk of stroke of peripheral vascular and/or cardiogenic origin. When these three risk factors were included into regression model, patients with ≥2 vessels treated with prior PCI still had a significantly higher risk to develop stroke after CABG (SHR 2.199, 95%CI 1.068-4.530). Still history of stroke or transient ischemic attack (SHR 2.226, 9%CI 1.191-4.159) and history of atrial fibrillation (SHR 1.941, 95%CI 1.085-3.461), but not depressed left ventricular ejection fraction, were independent predictors of stroke. However, we do not have data on the true nature and the underlying causes of stroke”.
Reviewer 2 Report
Biancari et al., have evaluated the risk of late mortality and major adverse cardiovascular and cerebral events after CABG in patients with prior PCI. The study has been well designed and executed. However, there are a few technical concerns which limits the quality of the manuscript.
Authors should change the title considering the present study's specification as the current title has been commonly used by same author and many others as well.
Authors should work on introduction to provide basic information, stating objectives and creating hypothesis from their speculations with respect to available recent literature.
Discussion needs to be expanded. Please discuss your results in detail according to available literature. Authors have missed a lot of literature.
Authors have mentioned a little about thrombosis. Did authors consider any antiplatelet therapy in such cases (before or after the intervention)? What about blood viscosity and Blood pressure? Platelet count? Did authors consider these important parameters which can greatly affect this study? Please discuss in detail.
The topic is quite similar with other available studies with a little difference, including authors own previous studies. Please clearly state the novelty of your study. Also, mention the limitations of the study.
Author Response
1. Authors should change the title considering the present study's specification as the current title has been commonly used by same author and many others as well.
Response: we do agree with the Reviewer and we are grateful for this wise suggestion. We change the title and we do hope that the Editor and Reviewer accept the present one.
Changes: we changed the title of the article as follows: “Late Myocardial Infarction and Repeat Revascularization After Coronary Artery Bypass Grafting in Patients with Prior Percutaneous Coronary Intervention”.
2. Authors should work on introduction to provide basic information, stating objectives and creating hypothesis from their speculations with respect to available recent literature. Discussion needs to be expanded. Please discuss your results in detail according to available literature. Authors have missed a lot of literature.
Response: We thank the Reviewer for these comments. We respectfully disagree with the Reviewer on the need to comment on previous studies, because data on adjusted estimates of late mortality and MACCE is very limited as we stated. Indeed, the lack of such data is the reason why we performed the present analysis.
Changes: We added the following comment and related references on the scarcity of data on late outcome after CABG in patients with prior PCI: “...only a few studies reported on the adjusted estimates of late non-fatal events after CABG in patients with prior PCI (8,9,10,11). These studies did not employ competing risk method and non-fatal ischemic events were poorly defined. Similarly, only a very few studies performed multivariable analyses on the impact of prior PCI on late mortality (6,12,13).”.
3. Authors have mentioned a little about thrombosis. Did authors consider any antiplatelet therapy in such cases (before or after the intervention)? What about blood viscosity and Blood pressure? Platelet count? Did authors consider these important parameters which can greatly affect this study? Please discuss in detail.
Response: We appreciated the Reviewer’s comments of strategies preventing and potential causes of thrombotic events. We performed further analyses employing a number of covariates including antithrombotic drugs before surgery and at discharge as well as low platelets count. Only the use of ticagrelor or clopidogrel at discharge had a significant impact on the risk of repeat revascularization. However, we noted that significant collinearity occurred because patients with prior PCI were more frequently received antithrombotic drugs before and after surgery thank patients with prior PCI. Therefore, adding these covariates to the regression models does not guarantee reliable results. We did not add the results of these analyses, but certainly we are willing to provide the Editor and the Reviewers with this data in case they want to check these additional results.
Changes: A comment on this issue has been added to the Limitations section.
4. The topic is quite similar with other available studies with a little difference, including authors own previous studies. Please clearly state the novelty of your study. Also, mention the limitations of the study.
Response: We fully agree with the Reviewer on the importance to underline the novelty of these observed data. Accordingly, we added a short comment on this issue. We have previously described the limitations of study and we added a comment on the antithrombotic therapy of these patients. Our previous studies have dealt only with early outcomes, while this study focuses on 5-year outcomes whose data was not yet available at the time of our first analyses.
Changes: We added the following comment to the Discussion: “The novelty of this analysis resides in the information provided on major adverse cardiovascular events at 5-year follow-up in a rather large series of all-comers population”.
Round 2
Reviewer 2 Report
Authors have improvised the manuscript